# Rapid *Salmonella* Serovar Classification Using AI-Enabled Hyperspectral Microscopy with Enhanced Data Preprocessing and Multimodal Fusion

**DOI:** 10.3390/foods14152737

**Published:** 2025-08-05

**Authors:** MeiLi Papa, Siddhartha Bhattacharya, Bosoon Park, Jiyoon Yi

**Affiliations:** 1Biosystems and Agricultural Engineering, Michigan State University, East Lansing, MI 48824, USA; papameil@msu.edu; 2Computer Science and Engineering, Michigan State University, East Lansing, MI 48824, USA; bhatta70@msu.edu; 3United States Department of Agriculture, Agricultural Research Service, U.S. National Poultry Research Center, Athens, GA 30605, USA; bosoon.parkdw@gmail.com

**Keywords:** *Salmonella* serovar, hyperspectral microscopy, artificial intelligence, data preprocessing, multimodal fusion

## Abstract

*Salmonella* serovar identification typically requires multiple enrichment steps using selective media, consuming considerable time and resources. This study presents a rapid, culture-independent method leveraging artificial intelligence (AI) to classify *Salmonella* serovars from rich hyperspectral microscopy data. Five serovars (Enteritidis, Infantis, Kentucky, Johannesburg, 4,[5],12:i:-) were analyzed from samples prepared using only sterilized de-ionized water. Hyperspectral data cubes were collected to generate single-cell spectra and RGB composite images representing the full microscopy field. Data analysis involved two parallel branches followed by multimodal fusion. The spectral branch compared manual feature selection with data-driven feature extraction via principal component analysis (PCA), followed by classification using conventional machine learning models (i.e., *k*-nearest neighbors, support vector machine, random forest, and multilayer perceptron). The image branch employed a convolutional neural network (CNN) to extract spatial features directly from images without predefined morphological descriptors. Using PCA-derived spectral features, the highest performing machine learning model achieved 81.1% accuracy, outperforming manual feature selection. CNN-based classification using image features alone yielded lower accuracy (57.3%) in this serovar-level discrimination. In contrast, a multimodal fusion model combining spectral and image features improved accuracy to 82.4% on the unseen test set while reducing overfitting on the train set. This study demonstrates that AI-enabled hyperspectral microscopy with multimodal fusion can streamline *Salmonella* serovar identification workflows.

## 1. Introduction

*Salmonella* is one of the most significant foodborne pathogens, with serovars that exhibit distinct host reservoirs, pathogenicity, and epidemiological profiles, making accurate serovar-level discrimination essential [1,2]. The current standard for the detection and identification of *Salmonella* serovars in food relies on a multi-step culture-based method outlined in regulatory guidelines such as the Food and Drug Administration’s (FDA) Bacteriological Analytical Manual (BAM). This method involves an initial 24-h non-selective pre-enrichment, followed by a 24-h selective enrichment and subsequent isolation on selective agar media for an additional 24–48 h. Confirmatory serotyping via slide agglutination using O and H antigen-specific antisera (the Kauffmann–White scheme) further extended the identification process by 2–3 days to the result [3,4,5]. Efforts to reduce the overall processing times led to the development of rapid molecular detection methods, which partially minimized the need for selective enrichment [6,7,8]. However, these methods still require an initial enrichment step, may detect non-viable cells, and continue to necessitate pure isolates for definitive serovar identification [5]. Additionally, robust data analysis often involves specialized domain expertise and skilled personnel. Therefore, there remains an unmet need for a rapid one-step method capable of simultaneously detecting and identifying viable *Salmonella* serovars without extensive enrichment, thereby complementing and enhancing rapid screening approaches.

To address these unmet needs, recent advancements in optical imaging and artificial intelligence (AI) present promising solutions. Optical imaging techniques that leverage intrinsic bacterial characteristics such as scattering patterns, autofluorescence, and refractive index have allowed for label-free analysis, minimizing extensive sample preparation and reducing dependence on selective enrichment process [9,10,11]. Similarly, classical microscopy-based approaches were enhanced by image processing algorithms that differentiated bacterial species based on observable parameters such as shape, size, and refractive index [12]. More recently, incorporating deep convolutional neural networks (CNNs) has accelerated the analysis of high-resolution image data by leveraging convolutional filters to extract fine-grained image-based features and hierarchical relationships, thus enabling the simultaneous detection and identification of viable bacteria [13,14]. The integration of AI models underscores the value of spatial relationships within image data, highlighting hidden yet informative patterns that can enhance rapid classification of pathogens.

Hyperspectral microscopy provides an advanced optical imaging by capturing high-dimensional data across a broader electromagnetic spectrum, significantly enhancing pathogen classification capabilities. This non-invasive technique utilizes intrinsic bacterial light-scattering fingerprints, offering detailed insights into biochemical, metabolic, and structural properties beyond basic cellular morphology [15]. Historically, hyperspectral microscopy applications primarily focused on spectral data for pathogen classification, using image-based data mainly for identifying cellular regions of interest (ROIs) [15,16,17]. These studies employed various data preprocessing methods, followed by comparative evaluations of traditional machine learning classifiers, such as random forest (RF), support vector machine (SVM), and *k*-nearest neighbors (*k*-NN), to determine an optimal algorithm. Early studies demonstrated successful differentiation of closely related *Salmonella* serovars [18,19], yet their emphasis on model optimization, rather than end-to-end automation, necessitated manual preprocessing steps for data quality and interpretation. Manual feature selection was commonly used in early hyperspectral analysis, when computational limitations often required simplifying data through averaging or selecting representative wavebands. More recent data preprocessing approaches, such as data-driven feature extraction, may better support scalable workflows by reducing the need for manual input. More recent advancements in AI-enabled hyperspectral microscopy have effectively integrated spatial features through CNNs, enabling automated feature extraction and hierarchical modeling. These improvements have enhanced classification accuracy at the species level [20] and facilitated the discrimination of physiological states such as viability [21] or stress responses [22]. Furthermore, some of these studies have shown that the multimodal fusion of spectral and spatial data (e.g., morphology descriptors) improves classification performance. Given these promising advancements in other pathogens, applying multimodal fusion specifically to *Salmonella* serovar classification represents a critical next step toward achieving rapid, accurate, and generalizable serovar-level identification.

Thus, this study introduces a rapid AI-enabled hyperspectral microscopy method for the simultaneous detection and identification of *Salmonella* serovars with enhanced generalizability. The objectives of this study are to (i) develop a rapid serovar classification approach requiring minimal sample preparation without enrichment, (ii) compare manual feature selection and data-driven feature extraction approaches to optimize spectral data preprocessing, and (iii) evaluate the multimodal fusion of spectral and image-based features to enhance classification accuracy and reduce overfitting. As illustrated in Figure 1, our AI-enabled hyperspectral microscopy method addressed these objectives by directly analyzing fresh overnight cultures of five *Salmonella* serovars (Kentucky, Infantis, Enteritidis, 4,[5],12:i:-, and Johannesburg) without prolonged enrichment and extensive reagents. These serovars were selected based on their prevalence in outbreaks and the challenges they present for serovar-level discrimination using standard methods. According to recent CDC reports, Enteritidis and the monophasic variant 4,[5],12:i:- rank among the top five serovars associated with foodborne illness and hospitalization in the United States [23]. Kentucky and Infantis have been increasingly linked to poultry and are notable for the global emergence of antimicrobial-resistant clones [24]. Johannesburg has also been repeatedly associated with poultry-related outbreaks, often co-occurring with Enteritidis and Infantis [24,25]. To leverage the rich information present in hyperspectral data cubes, the analysis employed two parallel branches: a spectral branch and an image-based branch. Within the spectral branch, single-cell spectra derived from segmented cells were preprocessed using manual feature selection or data-driven feature extraction approaches, followed by utilizing traditional machine learning algorithms to determine the optimal combination of preprocessing method and classification algorithm. In the image branch, microscopy images (i.e., RGB composites representing the full microscopy field) were analyzed using a CNN architecture. Outputs from both branches were then integrated using prediction-level multimodal fusion. Overall, this study not only accelerates serovar-level detection and identification but also establishes the groundwork for automated end-to-end pathogen classification, significantly enhancing food safety monitoring and enabling timely interventions.

## 2. Materials and Methods

### 2.1. Data Acquisition

#### 2.1.1. Bacterial Strains and Sample Preparation

In this study, a total of five *Salmonella* serovars (Enteritidis, Infantis, Kentucky, Johannesburg, and 4,[5],12:i:-) were selected due to their critical roles in food safety and public health, and they were obtained from the Food Safety Laboratory at Cornell University. The *Salmonella* stock cultures were maintained at 4 °C on LB Lennox agar plates (Thermo Fisher, Waltham, MA, USA). Fresh cultures for data collection were inoculated into 9 mL of tryptic soy broth (Difco, BD, Sparks, MD, USA) supplemented with 0.6% (*w*/*v*) yeast extract (Difco, BD) and incubated at 37 °C for 16 h. Cells in the logarithmic phase were harvested by centrifugation at 4450 rpm for 15 min at 4 °C (SYTO 9, SLA-1500, Hampton, NH, USA) to ensure physiological consistency across samples, following previously published protocols for hyperspectral imaging of bacteria [22]. The final pellet was suspended in sterile de-ionized (DI) water for a final concentration of 10^5^ CFU/mL for hyperspectral microscopy image acquisition. The microscope slides of bacterial cells were prepared following a modified version of a previously published method [22]. For each serovar, 2 µL of bacterial suspension was deposited onto a sterile glass microscope slide and air-dried for 15 min at 23 °C with an average relative humidity of 43% in a fume hood. After 15 min, 1 µL of sterile DI water was added to affix a plastic coverslip, securing the cells in place, as depicted in Figure 2.

#### 2.1.2. Hyperspectral Microscopy and Data Consistency Check

Hyperspectral microscopy data collection was conducted for all five serovars following our previously published method [22]. A total of 500 hyperspectral data cubes (100 for each serovar) were acquired using an Olympus BX43 upright optical microscope (Evident Scientific, Waltham, MA, USA) equipped with CytoViva-patented enhanced darkfield illumination optics, a tungsten halogen lamp (CytoViva, Auburn, AL, USA), and a HinaLea 4250 hyperspectral camera (HinaLea, Emeryville, CA, USA). The spatial resolution of the imaging system was 1936 × 1216 pixels, covering a spectral range of 399–1000 nm with 303 spectral wavebands with 2-nm bandwidth. Data collection was performed using a 100× oil immersion objective lens (Olympus UPLFLN 100×) with an exposure time of 200 ms and a gain setting of 16 dB; these parameters were obtained by preliminary experiments for optimizing high quality in hyperspectral data cubes.

For each data collection session, three subcultures per serovar were incubated across two independent sessions to yield a total of six biological replicates. Each subculture was allocated 3 microscope slides for data collection, serving as technical replicates. For each technical replicate, 6 distinct fields of view (FOV) were imaged per microscope slide, with careful selection to prevent overlapping cells. This approach yielded a total of 500 hyperspectral data cubes (100 per serovar) for further analysis. Raw hyperspectral data were analyzed with the ENVI software (version 6.0, NV5 Geospatial) to ensure consistency between the dataset within each serovar.

### 2.2. Data Preprocessing and Spectral Feature Engineering

After confirming data consistency, raw hyperspectral data were processed following the workflow illustrated in Figure 1. The dataset was initially partitioned into training (70%) and testing sets (30%) (fixed random state = 42 for replication) to prevent data leakage during preprocessing and ensure reproducibility. Spectral information from bacterial cells were obtained following previously published methods. Briefly, an automated segmentation of single-cell regions of interest (ROIs) was performed using attention-gated residual U-Net (ARG2U-Net) [26] to derive mean single-cell spectra. The resulting spectra were then transformed using standard normal variate (SNV) to remove baseline effects caused by imaging conditions [27].

These mean spectra served as inputs in the spectral branch for subsequent comparisons between two feature engineering approaches: (i) manual feature selection and (ii) data-driven feature extraction. For manual feature selection, three characteristic wavebands corresponding to the most intense regions near the red, green, and blue ranges across all serovars were manually selected by following the methods in our previous study [22]. This approach reflects traditional practices in hyperspectral analysis, which often involve selecting key wavebands based on visual inspection or spectral peaks to reduce computational demands. Here, manual selection provided a human-interpretable baseline against which automated, data-driven methods could be compared. For data-driven feature extraction, principal component analysis (PCA) was applied to the training set to reduce dimensionality by capturing the most informative linear combinations across all wavebands, as opposed to manually selecting only three wavebands. PCA was implemented using the singular value decomposition function from the Python (version 3.8) library NumPy. Before PCA, the training data were mean-centered, and the same mean value was used for test data transformation. The number of principal components (PCs) was chosen to retain 99% of the original variance, ensuring minimal information loss and data redundancy. The PCA transformation was fitted exclusively on the training set and subsequently applied to the test set.

### 2.3. Machine Learning Models for Spectral Branch

#### 2.3.1. Model Architectures

Four supervised machine learning algorithms were implemented in the spectral branch using Python scikit-learn library to determine the best performing model for *Salmonella* serovar classification. These included three traditional machine learning models widely used in food safety research—*k*-NN, RF, and SVM [16,28,29,30,31,32,33,34]—as well as a shallow multilayer perceptron (MLP) model. *k*-NN uses a distance-based metric to classify data points based on their nearest neighbors. RF employs an ensemble approach, combining multiple decision trees to improve classification accuracy and reduce overfitting. SVM classifies samples by identifying an optimal hyperplane to maximize the margin between different classes in feature space. The shallow MLP model was designed to include one hidden layer containing 100 neurons with a rectified linear unit (ReLU) activation function to introduce non-linearity, followed by a linear output layer.

#### 2.3.2. Model Training

All machine learning models were trained using the PyTorch (version 2.4.1) library for Python [35]. Hyperparameter tuning for each model was performed via grid search and 10-fold cross-validation. For *k*-NN, hyperparameter tuning optimized the number of neighbors (*k*), leaf size, and distance metric (*p*), with a search space including *k* values from 1 to 4, leaf sizes from 1 to 50, and distance metrics of 1 and 2. For RF, hyperparameters optimized included tree depth, minimum samples per split, minimum samples per leaf, and the number of features considered for splitting. The search space included maximum depths of 10, 25, or unlimited, minimum samples per split (5, 7), and leaf sizes (3, 5), with the number of estimators fixed at 100. SVM was optimized with a search space that included a fixed regularization parameter (*C* = 1), a radial basis function kernel, and polynomial kernel degrees of 3 and 4. Model selection was based on classification accuracy. MLP was trained for 200 epochs using the Adam optimizer (initial learning rate = 0.001) and cross-entropy loss.

### 2.4. Multimodal Spectral-Spatial Fusion

#### 2.4.1. CNN for Image Branch

In the image branch, microscopy images (i.e., RGB composites representing the full microscopy field) were analyzed using a CNN to capture spatial context without relying on predefined morphological descriptors. The microscopy images were created by combining data points at wavelengths of 620 nm (red), 550 nm (green), and 450 nm (blue). Building on our previously published method [22], a modified EfficientNetV2 architecture was trained using the PyTorch Lightning deep learning framework [36]. The pretrained EfficientNetV2 variant (‘efficientnetv2_rw_s’) was fine-tuned on the training dataset using a cross-entropy loss function with a total batch size of 16. All layers of the model were unfrozen, allowing the network to fully adapt pretrained ImageNet-1k features to our domain-specific data. Training was conducted on 350 RGB composite images (70 per serovar) derived from a 70/30 train/test split of the 500 total hyperspectral data cubes, as detailed in Section 2.2. Various combinations of standard augmentation strategies from the Albumentations library [37] were applied to enhance model robustness. Optimization was performed using the AdamW optimizer with an initial learning rate of 0.0001 and a weight decay of 0.001, while a step-based learning rate scheduler reduced the learning rate by a factor of 0.3 every 10 epochs. To mitigate overfitting, early stopping was implemented based on validation loss monitored throughout each epoch.

#### 2.4.2. Fusion of Spectral and Image-Based Prediction Outputs

The multimodal fusion model concatenated prediction-level outputs from the best-performing spectral machine learning model and the CNN image classification model, as shown in Figure 1. Note that the output dimensions from the spectral and image branches differ: the spectral branch outputs a 32-dimensional vector derived from single-cell spectra, while the original EfficientNetV2 typically outputs a class prediction vector. To align these for multimodal fusion, we replaced the final classification head of EfficientNetV2 with a fully connected linear layer that outputs a 32-dimensional feature vector. This enabled prediction-level concatenation with the spectral features prior to final classification. The resulting spectral-spatial fused vector had a length of 64. This vector was then fed into the final classifier head, a feed-forward neural network consisting of one hidden layer with 64 neurons, followed by an output layer with 5 neurons corresponding to the number of *Salmonella* serovar classes. The fusion model was trained for 50 epochs using an AdamW optimizer, with different learning rates for the image (0.0001) and spectral (0.0001) branches and the fusion classifier head (0.001) to balance learning. The batch size, weight decay, and the remaining training procedures and hyperparameters were the same as in Section 2.4.1.

### 2.5. Model Evaluation and Performance Metrics

Each model’s performance was assessed using classification accuracy, precision, and recall, ensuring a comprehensive evaluation. The formulas used for evaluation were as follows:
(1)$$Accuracy=\;\frac{TP+TN}{TP+TN+FP+FN}\;\times 100$$
(2)$$Precision=\frac{TP}{TP+FP}\times 100$$
(3)$$Recall=\frac{TP}{TP+FN}\times 100$$ where TP, TN, FP, FN are true positive, true negative, false positive, and false negative, respectively.

## 3. Results

### 3.1. Comparision of Hyperspectral Data of Salmonella Serovars

The results in Figure 3 illustrate image-based and spectral data derived from the raw hyperspectral data cubes. Figure 3A shows example microscopy images of individual *Salmonella* cells cropped from the full microscopy field to illustrate the representative morphology for each serovar. Spatially, bacterial cells from all serovars appeared visually similar to the human eye. However, Figure 3B reveals clear spectral distinction by the mean single-cell spectra obtained from segmented bacterial cells within 100 hyperspectral data cubes per serovar. Each serovar showed characteristic variations in mean spectral intensity and peak wavelengths. Specifically, *Salmonella* Kentucky exhibited a mean maximum intensity of 848 at 570 nm, *Salmonella* Johannesburg 1095 at 564 nm, *Salmonella* Enteritidis 812 at 562 nm, *Salmonella* Infantis 1191 at 555 nm, and *Salmonella* 4,[5],12:i:- 1159 at 553 nm.

### 3.2. Selection of the Optimal Classification Model Within the Spectral Branch

#### 3.2.1. Influences of Manual Feature Selection and Data-Driven Feature Extraction

Global manual feature selection was performed on the mean single-cell spectra to identify characteristic wavebands useful for *Salmonella* serovar classification. By analyzing these spectra, three characteristic spectral wavebands at 499, 555, and 628 nm were identified. These wavebands exhibited significantly higher intensity levels compared to those typically captured by wavebands represented by standard red, green, blue channels. This observation highlights the potential importance of these specific wavebands for differentiating among *Salmonella* serovars. Additionally, data-driven feature extraction using PCA was performed on the single-cell spectral dataset. It was found that retaining 18 principal components (PCs) was sufficient to explain 99% of the total data variance, as illustrated in Figure 4, which shows the cumulative variance explained by each PC. By selecting these 18 PCs, dimensionality was effectively reduced without the loss of critical information needed for accurate classification.

#### 3.2.2. Performance Comparison of Machine Learning Models Using Spectral Features

To identify the highest performing model within the spectral branch, the classification performance of machine learning algorithms was compared using both manually selected spectral wavebands and PCA-derived spectral features. Table 1 summarizes the classification performance obtained from each approach using three widely accepted metrics: accuracy, precision, and recall. Accuracy provides an overall measure of correct predictions (Equation (1)), precision quantifies the proportion of correct positive predictions (Equation (2)), and recall reflects the ability to detect all relevant instances (Equation (3)). In biological classification tasks, model accuracy is commonly interpreted in context. Values above 80% are generally considered strong and suitable for real-world applications. Accuracy above 90% typically indicates low-noise, well-separated class distributions, while values below 70% are often considered weak unless justified by factors such as class imbalance, high data variability, or exploratory objectives. Precision and recall offer complementary insights beyond accuracy. Values above 80% for both are also considered strong, indicating few false positives (high precision) and few false negatives (high recall). When employing the three manually selected wavebands, classification accuracies on the test sets (*n* = 150) for *k*-NN, SVM, RF, and MLP were 60.1%, 54.1%, 59.5%, and 62.2%, respectively, with corresponding precision values of 60.1%, 56.0%, 58.8%, and 62.1%, respectively, and recall values of 60.1%, 54.1%, 59.5%, and 62.2%, respectively. In contrast, the use of 18 PCA-derived spectral features, which captured 99% of the total data variance, substantially improved the performance of all models. Specifically, test-set classification accuracies increased to 73.7%, 75.0%, 77.7%, and 81.1% for *k*-NN, SVM, RF, and MLP, respectively, along with improved precision values of 74.2%, 76.3%, 78.6%, and 81.1%, respectively, and recall values of 73.7%, 75.0%, 77.7%, and 81.1%, respectively. From these results, the data-driven feature extraction with the MLP model (i.e., PCA-MLP) performed the best in the spectral branch and was used for multimodal fusion. This model achieved 81.1% test accuracy with equally high precision and recall, indicating consistent performance across classes without over- or under-prediction. These results support the effectiveness of data-driven feature extraction for *Salmonella* serovar classification.

Additionally, comparing performance on train and test sets provides insight into model generalization. Ideally, a well-performing model should show high accuracy on both sets with only a modest drop from train to test. A large discrepancy often indicates overfitting, where the model performs well on seen data but poorly on unseen data. Conversely, if both scores are low, it may suggest underfitting or limited model capacity. As shown in Table 1, RF achieved 100% accuracy on the train set but substantially lower accuracy on the test set (59.5–77.7%), suggesting overfitting. In contrast, PCA-MLP maintained high performance on both the train (99.4%) and test (81.1%) sets with balanced precision and recall, indicating strong generalization and robust classification.

### 3.3. Multimodal Classification by Fusion of Spectral and Image-Based Features

The multimodal fusion model, which combined the PCA-MLP spectral branch and a CNN-based image branch (EfficientNetV2), demonstrated superior classification performance compared to the standalone spectral and image-based models (Table 2). The PCA-MLP model refers to a shallow MLP trained on PCA-derived spectral features, which outperformed other machine learning models in the spectral branch (Table 1) and was therefore selected for multimodal integration. The CNN-based image classification model alone (EfficientNetV2) achieved a test-set accuracy, precision, and recall of 57.3%, indicating limited performance when using spatial features alone. In contrast, the spectral-only model (i.e., PCA-MLP) achieved an accuracy of 81.1%, highlighting the stronger predictive power of spectral data. The multimodal approach surpassed both standalone models, achieving a test-set classification accuracy, prevision, and recall of 82.4%. By integrating spectral and image-based features, the multimodal fusion reduced classification ambiguity, especially among serovars with overlapping spatial characteristics or subtle biochemical differences. Additionally, it substantially reduced overfitting observed in standalone models, narrowing the performance gap between training and test datasets (e.g., spectral-only model: 99.4% train vs. 81.1% test, multimodal model: 93.6% train vs. 82.4% test). This demonstrates improved generalization and highlights the advantage of combining complementary spectral and image-based features.

Serovar-specific classification performance is further illustrated in Figure 5. The multimodal model performed particularly well for the serovar 4,[5],12:i:-, achieving the highest classification accuracy of 97%. The lowest accuracy was observed for Infantis (73%), which was commonly misclassified as Johannesburg. Enteritidis (81%) and Johannesburg (82%) showed strong classification results with minimal confusion. Kentucky achieved moderate accuracy (79%) but showed occasional misclassification with Enteritidis. These serovar-specific patterns emphasize the model’s strength as well as highlight areas requiring further refinement.

## 4. Discussion

### 4.1. Hyperspectral Microscopy Captures Intrinsic Differences Among Salmonella Serovars

Hyperspectral microscopy in this study enabled a comprehensive spectral and image-based characterization of *Salmonella* serovars by simultaneously capturing spectral signatures and pixel-level spatial information at the single-cell level. The mean single-cell spectral profiles exhibited distinct intensity patterns for each serovar (Figure 3) despite relatively small variability compared to the spectral differences reported across bacterial species or viability states in previous studies [21,22]. This limited inter-serovar variation reflects phenotypic similarities yet reveals subtle biochemical differences that can support classification. These observations align with previous studies using hyperspectral microscopy with different light sources, confirming that the captured features reflect intrinsic characteristics of *Salmonella* [19,38]. Eady and Park (2016b) reported three critical spectral peaks between 400–800 nm using a tungsten halogen lamp [38], which is consistent with our data (Figure 3), alongside three additional peaks at 446 nm, 555 nm, and 628 nm. These spectral differences can be attributed primarily to variations in surface components among serovars, including lipopolysaccharides (LPS), outer membrane proteins, and other envelope-associated structures [2,39]. In particular, differences in the structure and composition of LPS, especially O-antigen polysaccharides and core oligosaccharides, are well documented among *Salmonella* serovars and can alter surface physicochemical properties, thereby influencing their optical and spectral profiles [40]. Such molecular differences influence the way each serovar interacts with incident light, leading to distinct spectral signatures [41,42]. Additionally, the source of each serovar isolate, whether environmental, clinical, or food-related, may contribute to variations in surface chemistry and structure, thereby affecting spectral properties [43,44,45]. Follow-up studies using higher-resolution or molecularly specific techniques, such as transmission electron microscopy or Raman spectroscopy, could help validate and localize the structural differences underlying the observed spectral variations. Furthermore, expanding the spectral range to include shortwave infrared (SWIR) could potentially improve serovar discrimination by capturing biochemical constituents.

### 4.2. Data-Driven Feature Extraction Improves Spectral Data Representation

Our results demonstrate that data preprocessing plays a crucial role in optimizing hyperspectral microscopy data for accurate *Salmonella* serovar classification (Table 1). The high dimensionality of hyperspectral data introduces challenges such as noise, redundancy, and irrelevant variability, all of which can negatively impact AI model performance [33,46,47]. In our previous study, manual feature selection based on domain knowledge achieved high classification accuracy for *E. coli* under different physiological states [22]. However, applying this approach to serovar-level classification of *Salmonella* resulted in reduced performance (Table 1). This decline may be attributed to the heuristic selection of only three wavebands, which may have excluded subtle yet informative spectral cues critical for distinguishing among serovars. While manual feature selection offers advantages in computational efficiency and data storage, it introduces subjective bias and risks omitting spectral regions that, though individually weak, contribute collectively to classification performance. This can result in the loss of spectral richness necessary for fine-scale discrimination.

In contrast, data-driven feature extraction using PCA substantially improved model performance (Table 1). PCA transforms hyperspectral data by projecting them onto orthogonal components, yielding a reduced set of uncorrelated features that retain the most informative variance. This transformation not only reduces dimensionality but also enables downstream models to focus on latent spectral patterns that may not be apparent through manual selection based on individual wavebands [48]. The superior performance of PCA-preprocessed data highlights the effectiveness of data-driven feature extraction in preserving subtle and distributed spectral information critical for serovar-level discrimination. Supporting this observation, a previous study using multivariate data analysis of *Salmonella* serovars reported classification accuracies above 90% when PCA was applied to mean single-cell spectra [18]. While that study relied on traditional statistical modeling, our results extend these findings by demonstrating the value of PCA as a preprocessing step within AI-enabled classification frameworks. These results underscore the effectiveness of unsupervised data-driven approaches in capturing the spectral complexity required for high-resolution microbial classification.

These findings are consistent with other strategies designed to manage the high dimensionality of both micro- and macroscale hyperspectral data. For instance, one study reported improved classification performance and reduced computation time when using PCA-guided waveband selection [49]. Moreover, Tanskin et al. (2017) proposed a data processing approach that combined dimensionality reduction with hierarchical function decomposition, which outperformed conventional feature selection methods [50]. These examples reinforce a broader conclusion observed across the literature that analytical decisions during preprocessing directly impact model outcomes. Improper preprocessing can introduce artifacts or remove biologically meaningful information [51], and variations in steps like normalization, noise reduction, and baseline correction significantly influence downstream classification performance [52]. Therefore, selecting appropriate preprocessing methods is essential to ensure that results reflect true biological variation rather than artifacts of data handling.

### 4.3. Multimodal Fusion of Spectral and Image-Based Features Enhances Classification and Mitigates Overfitting

The enhanced performance of the multimodal fusion model highlights the value of integrating multiple feature types derived from hyperspectral microscopy (Table 2). Combining spectral and image-based modalities is particularly advantageous for classifying microorganisms with low interclass variability [53]. Spectral data, which capture how samples scatter light across a wide range of wavebands, encode information related to biochemical compositions [18,19]. These signatures are especially useful for differentiating *Salmonella* serovars that may exhibit similar morphology but differ in molecular characteristics. As such, spectral features alone can provide strong biochemical fingerprints for classification tasks [54,55]. However, image-based features add complementary value to spectral signatures. As a type of spatial information, these features capture structural patterns directly observable within hyperspectral data cubes at the single-cell level. In our study, spectral and image-based feature vectors were combined at the prediction level, enabling a more comprehensive representation of each *Salmonella* serovar. This multimodal integration improved classification accuracy from 57.3% (image-based only) and 81.1% (spectral only) to 82.4% while also mitigating overfitting, as indicated by the reduced gap between training and test performance metrics (Table 2).

Although the dataset consisted of 500 hyperspectral data cubes (100 per serovar), it was substantially smaller than typical datasets used for algorithm development in large-scale image classification, such as ImageNet-1k with over 1.2 million natural images across 1000 classes. However, this study focused on applying existing model architectures to a biologically rich dataset, where each hyperspectral cube captureed detailed spectral and spatial information at the single-cell level, offering a high-dimensional representation uniquely suited for microbial classification. By leveraging transfer learning with pretrained EfficientNetV2 weights, the image branch adapted general image features to our biological task without requiring model training from scratch. The optimal dataset size for hyperspectral microbial classification remains an open question, as it likely depends not only on sample number but also on data richness and biological complexity. Follow-up studies will expand both the number and diversity of serovars and isolates to evaluate generalizability across broader strain-level and environmental variation.

In addition to overall model performance, we also examined classification outcomes by serovar. Classification performance varied by serovar, with Infantis exhibiting the lowest accuracy and often being frequently misclassified as Johannesburg (Figure 5). This observed misclassification likely reflects their underlying biochemical and genetic complexity. Infantis exhibits substantial genomic diversity, including variations in antimicrobial resistance genes and plasmid content, which can lead to heterogeneous spectral signatures within this serovar [56,57]. Johannesburg, on the other hand, has distinct bacteriophage–host interactions linked to unique LPS structures that differentiate it from many serovars but may partially overlap with certain Infantis variants [58]. The diversity within Infantis and overlapping surface biochemistry with Johannesburg likely produce similar optical properties, leading to misclassification consistent with their partial phylogenetic and phenotypic relatedness, especially in LPS composition. Such variability highlights the importance of integrating complementary data modalities to improve classification robustness. While this study focused on classification using monocultures under controlled conditions, practical deployment will require adaptation to more complex sample types. Future studies will assess performance in mixed cultures and relevant food matrices, where background microflora and matrix components may influence spectral signatures. These conditions will help test the robustness and generalizability of our multimodal approach in real-world food safety contexts.

Our findings align with a broader body of work demonstrating that integrating spectral and spatial information can improve classification performance in hyperspectral imaging applications. A conceptually related approach combined spatial features derived from segmented cells with spectral features to classify *Bacillus megaterium* and *Bacillus cereus*, achieving classification accuracies above 95% using traditional machine learning algorithms [59]. Although this method did not employ prediction-level fusion, it demonstrates that integrating spatial and spectral features can significantly enhance classification performance. Additionally, several studies have explored CNN architectures to better exploit the rich spectral and spatial information embedded in hyperspectral imaging data [20,27,60]. One study employed feature fusion strategies to combine spectral and spatial representations extracted from hyperspectral data cubes, while another implemented prediction-level fusion by integrating morphological descriptors, single-channel images, and spectral features within a unified classification framework [20,27]. In macroscale applications, 3D CNNs have also been developed to jointly learn spatial–spectral patterns by treating hyperspectral data cubes as volumetric inputs [60]. These findings underscore the effectiveness of multimodal fusion for robust pathogen classification and support future studies aimed at enhancing generalization across diverse microbial strains and environmental contexts.

## 5. Conclusions

This study demonstrates a rapid *Salmonella* serovar classification approach that reduces the need for selective enrichment by integrating hyperspectral microscopy with AI-enabled analysis. Hyperspectral data cubes were processed to extract two complementary data types: single-cell spectral profiles obtained from segmented cells and microscopy images constructed as RGB composites of the full microscopy field. By capturing both spectral and image-based spatial features, this approach enables comprehensive characterization of serovar-level bacterial differences, leveraging both biochemical signatures and pixel-level spatial patterns. The results illustrate the critical role of data preprocessing, showing that data-driven feature extraction via PCA outperforms manual feature selection by effectively preserving informative spectral variance while reducing data dimensionality. Furthermore, the image branch employed a CNN to learn spatial features directly from images, without relying on explicit morphological descriptors. The multimodal fusion of spectral and image-based features not only enhanced classification performance but also mitigated overfitting, ensuring robust and generalizable serovar classification. In this context, rapid classification was achieved through the streamlined biological workflow that bypassed selective enrichment, enabling direct analysis of fresh overnight cultures using rich hyperspectral microscopy data and AI-enhanced analysis to capture intrinsic spectral signatures of *Salmonella* serovars. Overall, these results highlight the potential of AI-enabled hyperspectral microscopy as a rapid, culture-independent solution for pathogen identification and food safety monitoring.

## Figures and Tables

**Figure 1 foods-14-02737-f001:**
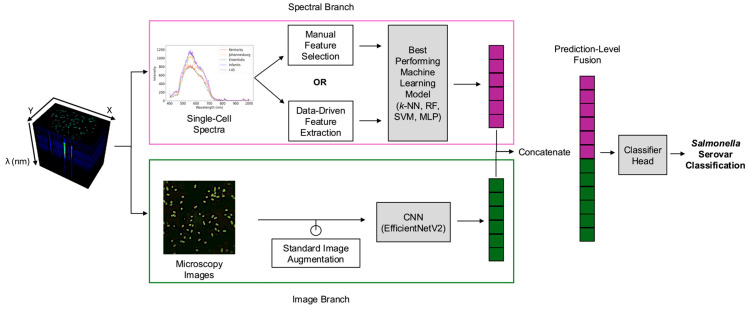
Overall workflow of hyperspectral data collection, preprocessing, and classification. *k*-NN: *k*-nearest neighbors. RF: Random Forest. SVM: Support vector machine. MLP: Multilayer perceptron. CNN: Convolutional neural network.

**Figure 2 foods-14-02737-f002:**
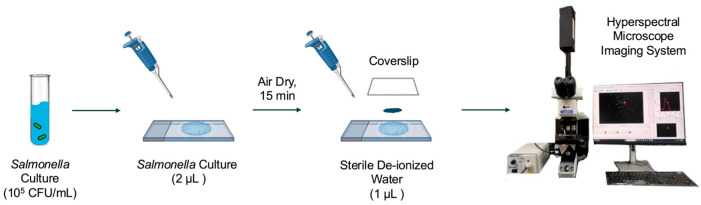
Bacterial sample preparation procedure for hyperspectral microscopy data acquisition.

**Figure 3 foods-14-02737-f003:**
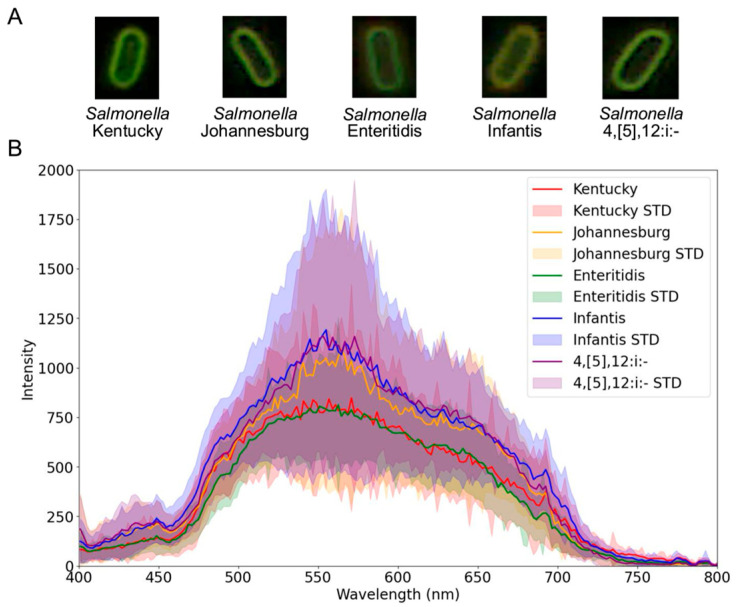
Hyperspectral data collected for *Salmonella* serovars: (**A**) Example microscopy images (i.e., RGB composites generated from selected hyperspectral wavebands: 449–451 nm [blue], 549–551 nm [green], and 619–621 nm [red]) showing one representative cell per serovar, and (**B**) mean single-cell spectral profiles with standard deviation (STD), derived from segmented cells across 100 hyperspectral data cubes per serovar.

**Figure 4 foods-14-02737-f004:**
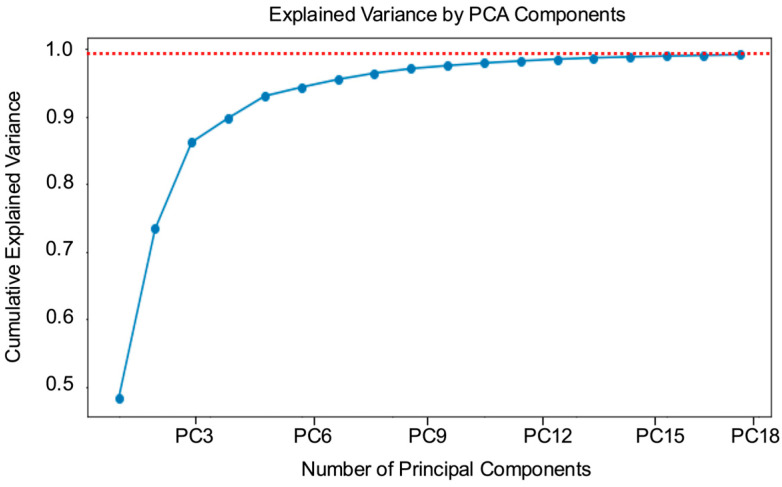
Principle component analysis (PCA) of spectral data from all *Salmonella* serovars. The cumulative variance explained by each principal component is shown. The first 18 principal components (PCs) accounted for 99% of the total variance in the dataset, as indicated by the red dotted line.

**Figure 5 foods-14-02737-f005:**
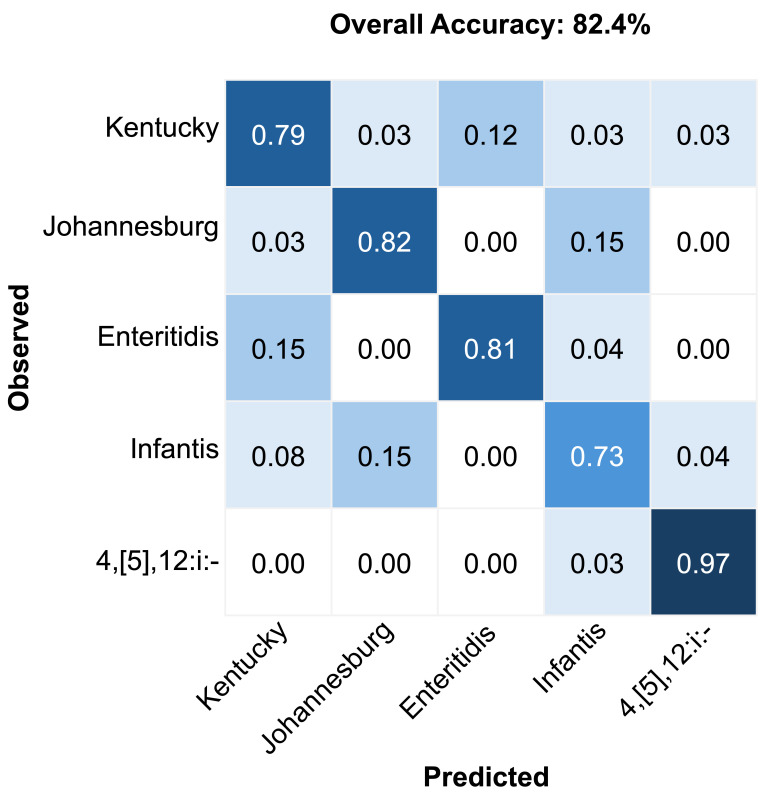
Confusion matrix results for the classification of *Salmonella* serovars using the multimodal fusion model.

**Table 1 foods-14-02737-t001:** Impact of feature engineering on spectral data with machine learning models using manual feature selection and data-driven feature extraction.

Spectral Features		Accuracy		Precision		Recall	
	Model	Train	Test	Train	Test	Train	Test
Manual selection ^1^	*k*-NN	67.2%	60.1%	67.7%	60.1%	67.2%	60.1%
	SVM	55.9%	54.1%	58.1%	56.0%	55.9%	54.1%
	RF	100.0%	59.5%	100.0%	58.8%	100.0%	59.5%
	**MLP**	**64.6%**	**62.2%**	**65.0%**	**62.1%**	**64.6%**	**62.2%**
Data-driven extraction ^2^	*k*-NN	81.7%	73.7%	81.9%	74.2%	81.7%	73.7%
	SVM	81.7%	75.0%	82.1%	76.3%	81.7%	75.0%
	RF	100.0%	77.7%	100.0%	78.6%	100.0%	77.7%
	**MLP**	**99.4%**	**81.1%**	**99.4%**	**81.1%**	**99.4%**	**81.1%**

^1^ The three characteristic spectral wavebands were manually selected based on domain knowledge, retaining physical interpretability. ^2^ The top 18 principal components (PCs) were extracted through principal component analysis, representing 99% of data variance. These PCs were linear combinations of all original spectral wavebands, creating abstract feature representations optimized for variance and reduced correlation or collinearity but lacking in direct physical interpretability. Bold values indicate the highest performance.

**Table 2 foods-14-02737-t002:** Enhanced classification through multimodal fusion of microscopy images with the best-performing model on single-cell spectra.

	Accuracy		Precision		Recall	
Model	Train	Test	Train	Test	Train	Test
Spectral only ^1^	99.4%	81.1%	99.4%	81.1%	99.4%	81.1%
Image only ^2^	68.2%	57.3%	77.8%	57.3%	77.8%	57.3%
**Multimodal fusion**	**93.6%**	**82.4%**	**86.4%**	**82.4%**	**86.4%**	**82.4%**

^1^ PCA-MLP was selected as the best model in Table 1. ^2^ EfficientNetV2 variant was used as the CNN model architecture. Bold values indicate the highest performance.

## Data Availability

The processed dataset (i.e., single-cell spectra and RGB composite images) is available on Zenodo at DOI:10.5281/zenodo.16740800 (https://zenodo.org/records/16740800, accessed on 4 August 2025). The code is available at https://github.com/food-ai-engineering-lab/salmonella-serovar-classification-foods (accessed on 4 August 2025), and future updates will be integrated into this repository.

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
