# Peer review of "Rapid Salmonella Serovar Classification Using AI-Enabled Hyperspectral Microscopy with Enhanced Data Preprocessing and Multimodal Fusion"

_foods, 2025, doi:10.3390/foods14152737_

Round 1
Reviewer 1 Report
Comments and Suggestions for Authors
The authors applied AI-enabled hyperspectral microscopy with enhanced data preprocessing and multimodal fusion to classify the Salmonella serovar. It was an interesting study. However, several issues must be addressed.
Comments:
- In the Abstract, the experimental details should be scaled down and added more description about the results.
- In the Introduction, past tense should be used, such as L44-50, L59-65, L78-85, L90-92.
- What is the basis for the manual selection of features?
- How to comprehensively evaluate the performance of the developed models? From Table 1, does the testing performance of data-driven extraction was acceptable?
- There were several studies about the classification of Salmonella serovar. Was the performance of the developed models in this work higher than others?
- From the Title, the author highlighted “Rapid classification”. Please give the training and testing time.
Author Response
Please see the attached response document. Thank you.

Reviewer 2 Report
Comments and Suggestions for Authors
The manuscript provided a method to identify and differentiate Salmonella serovars, which is novel. And the manuscript is well-written. I also have some comments for the manuscript. Please see details as follows.
The tested Salmonella serovar was prepared in sterile nutrient media. How does the method perform in differentiating serovars within a food matrix, which typically contains background microflora?
L29: The term “Multimodal MLP-CNN fusion” appears only in the abstract. Please clarify where and how the Multimodal MLP-CNN fusion was used in the main text, and ensure consistent usage throughout the manuscript.
L116-119 & L251-253: These sections contain text that pertains to journal submission requirements and should be removed from the manuscript.
L129: Why were logarithmic-phase cells specifically used in the study? Has the method been evaluated for its ability to detect Salmonella serovars in other growth phases, such as the lag or stationary phase?
L132: Please change ‘105 CFU/mL’ to ‘10^5 CFU/mL’.
L237: Please verify whether “AdamW optimizer’ was spelled correctly.
L262: The authors mention that n = 100 was used per serovar. Was the maximum intensity reported a mean value across replicates? Please clarify this point.
L267-268: Please specify the waveband used for capturing these microscopy images.
In section 3.2.1, the PCA analysis results were missing. Please complement.
In section 3.2.2, please clarify which model was identified with the best performance and used for the multimodal fusion model. Please include the relevant findings in the main text.
In the method-section 2.4.1, CNN was used to analyze the image branch; however, the corresponding results were missing. Please add.
L305: Please specify what model was used in the multimodal fusion model, besides CNN.
L321: Please define and explain what PCA-MLP refers to, and explain how it was selected in the main text based on Table 1.
References: The formatting of references is inconsistent, and there are typographical errors (e.g., Salmonella should be italicized). Please revise all references to meet the journal’s formatting guidelines.
Author Response

(The authors gave the same response as above.)

Reviewer 3 Report
Comments and Suggestions for Authors
The manuscript presents an innovative study on the rapid classification of Salmonella serovars using AI-integrated hyperspectral microscopy. The authors combine spectral and spatial information through multimodal fusion, achieving improved classification accuracy compared to previous methods. The topic is timely and relevant to food safety monitoring, aligning well with the scope of Foods. Overall, the study is well-structured, technically solid, and offers significant contributions.
However, certain aspects require clarification, additional detail, and moderate revision to improve the manuscript’s rigor and readability:
-The rationale for selecting only five Salmonella serovars should be better justified. Are these the most prevalent in foodborne outbreaks, or particularly difficult to discriminate? Please clarify in the Introduction.
-The dataset consists of 500 data cubes (100 per serovar). While sufficient for a proof-of-concept, it remains relatively small for training deep learning models like EfficientNetV2. The authors should discuss: Potential overfitting risk due to limited sample diversity. Plans for future studies with expanded datasets to validate generalizability.
-The manuscript states that data and code will be made available in a public repository upon publication. Please provide a DOI or a working link (currently just the GitHub homepage is listed).
-More specifics on the EfficientNetV2 modifications are needed. For instance:
- How was the model architecture modified to output a 32-dimensional vector instead of class predictions?
- Was the pretrained network fully fine-tuned, or were earlier layers frozen?
- How large was the effective training set for the CNN?
-The study highlights spectral differences among serovars, but biological explanations remain somewhat speculative. For example:
- Could authors relate specific spectral features to known biochemical differences among the serovars (e.g., differences in LPS composition)?
- Is there any correlation between misclassification rates (e.g., Infantis vs. Johannesburg) and known phylogenetic relationships?
Author Response

(The authors gave the same response as above.)

Round 2
Reviewer 1 Report
Comments and Suggestions for Authors
The authors had revised the manuscript based on the comments. It can be accepted now.
Reviewer 2 Report
Comments and Suggestions for Authors
Thank you for addressing the comments.
Reviewer 3 Report
Comments and Suggestions for Authors
From my side, the manuscript can be publisahble.